# Local Effects of Ring Topology Observed in Polymer Conformation and Dynamics by Neutron Scattering—A Review

**DOI:** 10.3390/polym12091884

**Published:** 2020-08-21

**Authors:** Valeria Arrighi, Julia S. Higgins

**Affiliations:** 1Institute of Chemical Sciences, School of Engineering and Physical Science, Heriot-Watt University, Edinburgh EH14 4AS, UK; 2Chemical Engineering Department, Imperial College London, South Kensington Campus, London SW7 2AZ, UK

**Keywords:** small angle neutron scattering, conformation, polymer topology, quasi-elastic neutron scattering, neutron spin-echo, segmental motion, cyclic polymers

## Abstract

The physical properties of polymers depend on a range of both structural and chemical parameters, and in particular, on molecular topology. Apparently simple changes such as joining chains at a point to form stars or simply joining the two ends to form a ring can profoundly alter molecular conformation and dynamics, and hence properties. Cyclic polymers, as they do not have free ends, represent the simplest model system where reptation is completely suppressed. As a consequence, there exists a considerable literature and several reviews focused on high molecular weight cyclics where long range dynamics described by the reptation model comes into play. However, this is only one area of interest. Consideration of the conformation and dynamics of rings and chains, and of their mixtures, over molecular weights ranging from tens of repeat units up to and beyond the onset of entanglements and in both solution and melts has provided a rich literature for theory and simulation. Experimental work, particularly neutron scattering, has been limited by the difficulty of synthesizing well-characterized ring samples, and deuterated analogues. Here in the context of the broader literature we review investigations of local conformation and dynamics of linear and cyclic polymers, concentrating on poly(dimethyl siloxane) (PDMS) and covering a wide range of generally less high molar masses. Experimental data from small angle neutron scattering (SANS) and quasi-elastic neutron scattering (QENS), including Neutron Spin Echo (NSE), are compared to theory and computational predictions.

## 1. Introduction

The physical properties of polymer chains are very sensitive to a range of both structural and chemical parameters. Chemical structure, molar mass, conformation, but also tacticity, chain architecture, and topology determine, for example, the extent of local interactions between polymer chains, glass transition and crystallization behavior, the degree of chain entanglement ultimately affecting solution properties, viscosity, and mechanical response [1,2]. Structural and dynamic studies as a function of polymer architecture and topology, e.g., cyclic, star, and hyperbranched polymers, dendrimers, and rotaxanes, are of considerable interest since these structures may display unique physical properties leading to new applications. In this review we focus on work carried out on cyclics (in the literature the terms cyclic and ring are used interchangeably and to reflect this we also use both terms throughout the review).

Cyclic polymers differ from linear chains by one single bond that links the chain ends. This apparently trivial topological constraint has a profound effect on many polymer properties [3]. For example, it has been shown to influence crystallization [4,5,6], thermal properties such as heat capacity [7] and glass transition [6,8,9,10,11], bulk viscosity [12,13,14], and diffusion coefficients [15,16,17,18]. Early theoretical studies focused on the effect of polymer topology on conformation and radii of gyration [19,20,21,22], glass transition and dynamics [23]. Extensive work has also been carried out using computer simulations (see for example references [24,25,26,27,28,29,30,31,32,33,34,35,36]).

The conformation and dynamics of a linear polymer chain is well understood and documented in reviews and text books [37,38,39], but there are still open questions as to the impact of chain architecture and topology on the physical properties of polymers and their mixtures with linear chains. Thanks to advances in polymer synthesis, systems with well-defined architecture and molar mass have become available making it possible to revisit, in some cases, experimental findings and test a range of theoretical predictions [40]. This has stimulated new investigations.

Scattering techniques offer a unique opportunity to gather experimental data on the effect of topology on polymer conformation, miscibility, and dynamics. In particular, Small Angle Neutron Scattering (SANS), quasielastic neutron scattering (QENS), and Neutron Spin Echo (NSE) spectroscopy cover length and time scales suitable to address important questions regarding size, conformation, and local and longer range polymer motions [41,42,43].

Studies of macroscopic properties such as bulk viscosity [12,13,14] and diffusion coefficients [15,16,17,18] have been well documented in the literature, but there has been limited work on dynamic behavior at a molecular and sub-molecular level. Understanding such motion, is of great importance as it impacts on both mechanical and rheological properties. Recent reviews of the effect of topology on polymer chain dynamics (e.g., Richter et al. [44]) have a particular focus on relatively long-range molecular motion described by the Doi Edwards reptation model. The absence of chain ends in cyclic polymers means they are a unique test of the tunneling motion driving reptation. Such experimental tests require very pure fractions of high molecular weight cyclics, and, for use of the NSE technique, deuterated samples. Only relatively recently have such samples become available; however, going back several decades there has been a wealth of studies of much lower molecular weight cyclics, which has tended to escape the attention of the reviews. Here, we review the long history of theoretical and experimental studies on the effect of ring topology on conformation, local structure, and dynamics, as well as miscibility, putting in context of our own work carried out over the past 40 years using SANS and QENS neutron scattering experiments.

The review is organized as follows. In Section 2, we briefly summarize the main aspects of neutron scattering, both SANS and QENS. We then in Section 3 review SANS studies of cyclic polymers in solution and in bulk. Part 4 deals with the effect of topology on local dynamics. Section 5 gives an overview of work carried out on polymer mixtures with one component being a cyclic polymer, and finally, in Section 6, we summarize the current situation regarding understanding of the different behavior of ring and chain samples and their mixtures.

## 2. Neutron Scattering

Neutron scattering techniques rely on the interaction (scattering length) of neutrons with the nuclei of atoms within samples and this provides significant differences (and advantages) compared to X-ray or light scattering. Unlike X-rays, neutrons are particularly sensitive to light atoms. Furthermore, there exists a large difference between the scattering lengths, *b*, for hydrogen and deuterium (*b*_H_ = −3.739 × 10^−13^ cm and *b*_D_ = 6.671 × 10^−13^ cm, respectively). This special feature of neutron scattering has been used to a great advantage in many structural and dynamic studies of polymer systems [41]. Molecules or sections of molecules can be highlighted in experiments via isotopic substitution of D for H.

The spatial resolution of neutron experiments can range from angstroms to microns, allowing intra-molecular conformation and intermolecular structure to be explored in small angle neutron scattering experiments, SANS. The frequency range accessed by the neutron scattering technique (10^7^ to 10^14^ Hz) limits investigation to relatively high frequency molecular motions from local vibrations to diffusion. Two neutron experimental techniques have been applied to polymer dynamics, quasi-elastic scattering, QENS, and neutron spin echo, NSE [41]. A schematic representation of the physical quantities measured in QENS and NSE experiments and how these are related is given in Figure 1. We do not intend to discuss the details of these techniques, but a fundamental difference is important. QENS makes observations in the frequency domain through the dynamic structure factor, *S*(*Q*,*ω*), while the function describing the dynamics of a system is a time correlation function, *S*(*Q*,*t*). The QENS signal is the Fourier transform of the time correlation function. In NSE experiments, the time correlation function is measured directly. This difference has consequences both in removing resolution effects and in comparing experiment to theory. NSE tracks the coherent scattering which addresses the pair correlation function and has been particularly relevant for observing the movement of single chains in bulk samples [41]. QENS is more generally used for incoherent scattering where it is the self-correlation function that is observed. The scattering pattern is a sum over the motion of individual scattering centers and therefore, depending on temperature and chemical structure, several dynamic contributions may have to be considered.

While exploring a limited time scale of molecular motions, neutron scattering offers, through the changes in the wave vectors before and after scattering, a way of defining motion at specified distances. For QENS, the distance scale probed is limited to a few tens of angstroms, but NSE can reach distance above 100 Å. Combined, these techniques can be used to probe molecular motion and test predictions from theory or computational studies.

## 3. Cyclic Polymers

A comprehensive review of synthesis and properties of cyclic polymers has been published by Kricheldorf [3] and here we briefly highlight work, relevant to our discussion.

There are a number of ways of synthesizing large cyclic molecules, but in the literature two routes dominate. Ring-ring equilibration (e.g., for polysilanes and polisiloxanes) produces polydisperse cyclic polymers almost free from linear chains. Followed by preparative GPC fractionation, narrow molecular weight fractions of cyclics with molecular weights up to ca. 50,000 g mol^−1^ have been produced [45]. Alternatively, the end linking of highly monodisperse linear chains in dilute solution can produce very high molecular weight cyclics. Examples are polystyrene and polyethylene oxide. Both routes suffer from contamination of the ring samples by small amounts of linear chains. For the highest molecular weight rings the latter synthesis is the only possible route, but has the disadvantage of producing very small sample quantities.

Nevertheless, experiments on well characterized cyclic molecules have made it possible to confirm theoretical predictions of chain dimension in solution [19,21,46,47,48,49,50,51] and in bulk [52,53,54,55], leading to a good understanding of their structural properties.

Pioneering work on cyclic polymers, particularly cyclic poly(dimethylsiloxane) (PDMS), was undertaken by Semlyen and his co-workers starting from the 1960s [56]. By coupling ring-chain equilibria with preparative gel permeation chromatography (GPC), Semlyen’s group was able to isolate narrow fractions of cyclic PDMS with dispersity indices M_w_/M_n_ of 1.05 ± 0.02 and number-average molecular weights as high as 50,000 g mol^−1^, corresponding to number-average numbers of skeletal bonds up to 1300 [57]. Although investigations were later conducted on other cyclic systems [3], PDMS remains the most studied cyclic polymer, having been characterized by almost every available experimental technique from GPC [57] to rheology [12], NMR [17,18], dielectric spectroscopy [58], and small angle neutron scattering [46,52,53]. Other cyclic polymers studied to date include poly(styrene) (PS) [40,48,59,60], poly(butadiene) (PB) [61], and poly(ethylene oxide) (PEO) [44,62]. To aid discussion of the experimental data in the following sections, the chemical structures of the polymer repeat units and other relevant parameters are reported in Table 1.

For many years, due to a lack of deuterated cyclic samples, SANS studies of cyclics were limited to measurements of the radii of gyration [63] in solutions of a protonated polymer in a deuterated solvent. Recently, development of new HPLC separation techniques (e.g., liquid chromatography at the chromatographic critical condition, LCCC), together with progress in synthesis has improved availability of cyclic polymers and also of deuterated rings [40,64,65]. These advances have made it possible to compare microscopic structure and dynamics of rings in bulk samples to that of the linear chains. In the following sections we review our work on cyclic polymers in solution and in the undiluted state and discuss results in a more general context.

### 3.1. Conformation of Cyclic Polymers in Solution

The question of how the dimensions of cyclics compare to linear chains of the same chemical structure and degree of polymerisation has interested theoreticians and experimentalists over many years. One of the first studies to be reported in the literature is the work of Kramers [19,67], who proposed that the ratio of the mean-square radii of gyration of Gaussian chains, $R_{g,l}^{2}$, and ring polymers, $R_{g,c}^{2}$, at theta-conditions is a factor 2.

The first experimental study reporting a comparison between the average dimensions of cyclics and linear chains was published four decades ago by Higgins et al. [46]. Using SANS, the group determined the mean square radii of gyration $R_{g,c}^{2}$ of cyclic PDMS with z-average number of bonds between 130 and 550 in deuterated benzene solution and compared these values to those of PDMS linear chains, $R_{g,l}^{2}$.

Measurements of the neutron scattered intensity, *I*(*Q*) at small *Q* values (*Q* = 4π/λ sin(θ/2), where θ is the scattering angle and λ the neutron wavelength) give information on chain dimensions and polymer conformation. For a polymer solution, in the so-called Guinier range where *Q*
$R_{g}^{}$ ≤ 1, $R_{g}^{}$ values can be determined from the Zimm equation [41]:(1)$$\frac{Kc}{I\left(Q\right)}=\frac{1}{M}\left[1+\frac{Q^{2}R_{g}^{2}}{3}\right]+2A_{2}c$$
where *M* is the molecular weight of the polymer, *c* is the concentration and *A*_2_ represents the virial coefficient. Benzene is a good solvent for PDMS and so this additional term needs to be included to account for polymer-solvent interactions. The magnitude of the constant *K* is determined by the contrast:(2)$$K=\frac{{\left(\Delta \rho \right)}^{2}}{N_{A}\rho_{bulk}^{2}}$$
where *N*_A_ is Avogadro’s number and $\rho_{bulk}$ is the density. (Δ*ρ*)^2^ is the neutron contrast factor which is defined as the square of the difference between the neutron scattering length densities (SLD) of the scattering species and the matrix (e.g., the solvent or a different polymer) [41]. For a polydisperse system, the *M* and $R_{g}^{2}$ parameters in Equation (1) are replaced by the weight average molecular weight, *M*_w_ and the z-average mean-square radius of gyration, $\langle R_{z}^{2}\rangle$. The latter can be determined from the slope of a plot of the reciprocal of *I*(*Q*) versus *Q*^2^. According to Equation (1), for *Q* = 0, one could determine the virial coefficient *A*_2_ by plotting *Kc*/*I*(*Q*) versus concentration. Rather than generating different plots, for data collected at different concentrations, a double extrapolation procedure can be adopted to obtain all relevant parameters from the experimental data: *A*_2_, *M*_w_, and ${\langle R_{z,c=0}^{2}\rangle }^{1/2}$ (Zimm plot). As shown in Figure 2, SANS data at different concentrations are shifted along the x-axis by plotting *c*/*I*(*Q*) versus *Q*^2^ + *kc*, where *k* is a conveniently chosen constant.${\langle R_{z,c=0}^{2}\rangle }^{1/2}$ values are then obtained from the slope of the extrapolated line at *c* = 0, whereas *A*_2_ can be calculated from the slope of the extrapolated line at *Q*^2^ = 0. The intersection of the two extrapolated lines yields the weight average molar mass.

From similar measurements at different molar mass of the cyclics and linear chains, the ratio between $R_{g}^{2}$ values was found to be 1.9 ± 0.2. This is close to theoretical predictions for flexible polymers in the unperturbed state and consistent with the observation made by Kramers [67] many years ago that a decrease in the ratio is expected following expansion of rings and chains, in a thermodynamically good solvent such as benzene.

The scaling of chain dimensions in polymer solutions is represented by the Flory exponent, *ν*, which describes how polymer size, e.g., the radius of gyration, scales with molecular weight. As reported by us elsewhere [52], scaling exponents calculated from the SANS data of Higgins et al. [46] for cyclic and linear PDMS in benzene-d_6_ at 292 K are very similar, i.e., *ν* = 0.64 ± 0.02 and 0.68 ± 0.04, respectively, meaning that, in dilute solution, the same scaling law applies, irrespective of chain topology. This is in agreement with expectations based on the computer simulations of Muller et al. [31]. Thus, results on cyclic and linear polymers in solution indicate that the static properties of the rings are not altered by the presence of topological constraints and one expects the same swelling exponent for the two polymers.

Similar conclusions were reached for solutions of ring and linear polystyrene [50,68]. For example, recent SANS measurements by Takano et al. reported *ν* = 0.61 ± 0.02 and 0.60 ± 0.02 for linear and cyclic PS, respectively, in benzene-d_6_ [50]. The same authors, however, measured different scaling exponents for linear (*ν* = 0.49 ± 0.02) and cyclic (0.53 ± 0.02) PS in deuterated cyclohexane at 40 °C (a theta solvent for PS), a result that is consistent with work of Roovers [69]. It was argued that such a difference between *ν* exponents was significant and could be explained in terms of repulsive interactions between segments of the cyclic molecules at theta-conditions for linear polystyrene. This has been attributed to the “topological expansion effect” or the “topological swelling effect”, which was predicted by Deustch et al. [26], Grosberg et al. [70], Dobay et al. [71], Matsuda et al. [72], and Moore et al. [73]. The additional intrachain repulsion in rings can also account for their lower theta temperature and higher dissolution limits compared to the linear counterparts as shown by previous studies [51,70].

A detailed comparison of the form factor of pure, high molar mass PS rings (161,000 g mol^−1^), at theta- and good-solvent conditions, and in a dilute solution of linear PS chains by Gooßen et al. [74], demonstrated that ring dimensions increase with solvent quality. Interestingly, these authors showed that the experimental form factors in both theta-solvent and linear matrix can be superimposed and rings have nearly Gaussian conformations.

In some of the literature, authors have chosen to discuss the so-called the *g*-factor $\langle R_{g,c}^{2}\rangle /\langle R_{g,l}^{2}\rangle$ rather than the Flory exponent, *ν* of the rings or chains. As recently noted by Gartner et al. [65], for ideal Gaussian chains this ratio is expected to be equal to 0.5, but values ranging from 0.5 to 0.6 have been reported from both experiments and computations [50]. For PS samples with molecular weight from 17 to 570 k in benzene-d_6_, Takano et al. reported *g*-factors varying from 0.589 to 0.647. Lower values were determined by Ragnetti et al. [63] for PS in d-toluene (0.55), Lutz et al. [75] for PS in cyclohexane-d_12_ at theta temperature (0.53) and Higgins et al. [46] for PDMS in benzene-d_6_ (0.53).

Large variations between *g*-factors may be related to different solvent conditions, but sample purity is also believed to contribute to discrepancies between data reported by different groups. Gartner et al. [65] while coupling coarse-grained molecular dynamics simulations and SANS measurements on PS in d-cyclohexane studied the effect of solvent quality and concentrations on chain conformation, scaling, and interactions of linear chains and cyclics. Interestingly, these authors concluded that scaling is the result of a delicate balance between intra- and inter-chain contacts, and this is relatively insensitive to the ring closure chemistry and the presence of linear or other impurities.

### 3.2. Conformation of Cyclic Polymers in the Pure Melt State

Many theoretical and computational studies have been carried out on rings [22,24,25,31,33,76,77,78,79,80,81]. The general conclusion is that the conformational properties of rings in dilute solution and in the pure melt state (also referred to as bulk) differ significantly. Due to the lack of chain ends, unconcatenated, unknotted rings cannot adopt all possible conformations. Such restrictions give rise to an additional topological excluded-volume interaction between neighboring chains that is only negligible for dilute solutions.

Cates and Deutsch [22] were among the first to note that, due to the presence of topological constraints, cyclic molecules may adopt a more compact structure in bulk compared to linear chains. Thus, for any given number of segments N, rings in the undiluted state should not only be smaller but also partially collapsed compared to their linear counterpart. Using a Flory-like argument based on the free energy of a ring, the authors predicted a scaling law *R_g_* ∝ *N^ν^* with *ν* = 2/(d + 2), where d is the dimensionality and therefore for d = 3 one expects *ν* = 2/5 [22]. A scaling exponent equal to 0.39 ± 0.03 was reported by Muller, Wittmer, and Cates who carried out computer simulations on rings with up to 512 monomers and determined *ν* from finite-size analysis, in the limit of infinite *M* [31]. This result is in agreement with values found by other groups [24,25,76,78,82].

Therefore, rings in the undiluted state, i.e., in the presence of other rings, should adopt a collapsed conformation compared to the one expected based on Gaussian statistics [79,80,81]. These predictions can be tested experimentally by carrying out SANS experiments on mixtures of deuterated and hydrogenated cyclic molecules.

In the framework of the mean-field random phase approximation (RPA) derived by de Gennes [83] and Binder [84], the structure factor *S*(*Q*) for an H/D mixture of two chemically identical polymeric species is given by:(3)$$\frac{1}{S\left(Q\right)}=\frac{1}{\phi_{D}\nu_{D}N_{D}P_{D}\left(Q\right)}+\frac{1}{\phi_{H}\nu_{H}N_{H}P_{H}\left(Q\right)}-\frac{2\chi }{\nu_{o}}$$
where *N_i_* and *Φ_i_* are the degree of polymerization and volume fraction of polymer *i*, respectively, and *χ* is the segment-segment interaction parameter. The reference volume *ν_o_* is usually taken as the geometric average:(4)$$\nu_{o}=\sqrt{\nu_{H}\cdot \nu_{D}}$$

Since the difference between the volumes of hydrogenous and deuterated species is small, for a H/D blend, *ν_D_* is approximately equal to *ν_H_*. When accounting for the contrast, the Q dependence of the scattered intensity is given by [52]:(5)$$I\left(Q\right)=\frac{\left(1/\nu_{o}\right){\left(b_{H}-b_{D}\right)}^{2}}{{\left[\phi_{D}N_{D}P_{D}\left(Q\right)\right]}^{-1}+{\left[\phi_{H}N_{H}P_{H}\left(Q\right)\right]}^{-1}-2\chi }$$

Despite considerable theoretical and computational work having been carried out on the conformation of cyclic polymers, in bulk, experimental investigations became possible only once deuterated molecules became available. The first comparison between the conformation and thermodynamics of cyclic and linear polymers in bulk was reported for PDMS [52,53], thanks to work of Dagger and Semlyen who were able to prepare a range of per-deuterated cyclic PDMS fractions ranging from 45 to 605 number average of skeletal bonds [85,86].

For rings, the form factor *P*(*Q*) was calculated by Casassa, and it is given by [20]:(6)$$P\left(Q\right)=\left(\frac{2}{\sqrt{t}}\right)exp\left(-\frac{t}{4}\right)\int_{0}^{\frac{\sqrt{t}}{2}}exp\left(x^{2}\right)dx=\left(\frac{2}{\sqrt{t}}\right)D\left(\frac{\sqrt{t}}{2}\right)$$
where *t = Q^2^*$R_{g,c}^{2}$ and *D*(*x*) is the Dawson integral. This result was also confirmed by others [21,47].

As shown in Figure 3 for cyclic H/D PDMS mixtures [53], the Casassa form factor *P*(*Q*) [20] was found to better describe the *Q* dependence of the scattered intensity than the Debye equation [41]. This was certainly true for relatively small cyclics up to ca. 11,000 g mol^−1^ but the standard Debye equation had to be used in order to fit the SANS data of larger rings (*M*_w_ ca. 20,000 g mol^−1^). In a recent paper, Beaucage and Kulkarni [55] have refitted the data on PDMS rings and chains in references [52,53]. Using a dimensional analysis approach, which treats the cyclic polymers as mass fractal objects, they obtain a good fit of their unified model *P*(*Q*)to the data in [52,53] and extract a number of fractal parameters. They also discussed deficiencies in the Casassa model for values of *Q*$R_{g}^{}$ > 10. Fitting their model to recent simulation data, they see possible indications of chain collapse of higher molecular weight rings (not yet evident in experimental data).

Despite the limitations on the detailed fitting of *P*(*Q*) for the SANS data in references [52,53], data analysis made it possible to compare the weight-average radii of gyration of rings and linear chains in the undiluted state. As shown in Figure 4, both topologies follow a power law of type $R_{g,w}^{}$ ∝ *M^ν^* but cyclic molecules have smaller dimensions than predicted from Gaussian statistics. For linear PDMS chains, $R_{g,w}^{}$ ∝ *M*^0.53 ± 0.03^, which is close to the result predicted from Gaussian statistics. For cyclics with *M*_w_ < 11,000 g mol^−1^, $R_{g,w}^{}$ was found to scale with *M*^0.42 ± 0.05^ (Figure 4), in good agreement with computer simulations and theoretical predictions that chain dimension of cyclics in bulk should increase according to *N*^2/5^.

As pointed out by Pakula et al. [76], the ratio $R_{g,l}^{2}/R_{g,c}^{2}$ increases with increasing chain length, thus deviating from the prediction based on Gaussian statistics that $R_{g,l}^{2}/R_{g,c}^{2}$ = 2. Such a relationship is strictly valid only in dilute solution where topological interactions are absent. Data in Figure 4 clearly support this view.

SANS studies of cyclic polymers in bulk are limited to a few systems. In addition to the PDMS results discussed above, measurements have been carried out on hydrogenated and deuterated poly(ethylene oxide) (PEO) samples with high purity of cyclic product (>99.5%) [44,87,88] and, more recently, on PS samples [89] with *M*_w_ varying from 10,000 to 400,000 g mol^−1^. Similar to work on PDMS, for PEO rings up to *M*_w_ = 20,000 g mol^−1^ the mean-squared radius of gyration $R_{g}^{2}$ was found to scale with *N*^2^*^ν^* with *ν* = 0.43 ± 0.015 [44,54,87,88].

The presence of cyclic structures gives rise to a characteristic maximum in a Kratky representation, i.e., a plot of *I*(*Q*) *Q*^2^ versus *Q*, located at *Q R_g_*~2.05, based on numerical calculation of a ring form factor in a theta-solvent [74,90]. As shown in Figure 5 for a range of PEO samples, when data are plotted according to the Kratky representation and the x-axis scaled by *R_g_*, the maxima for different samples are located at roughly the same *Q*·*R_g_* value. For linear chains, the scattered intensity shows no peak in a Kratky plot but a plateau is observed at 12/$R_{g}^{2}$. Richter et al. [44] nicely demonstrated that for a cyclic molecule, the Kratky plateau of a ring should be suppressed by the ratio of the radius of gyration of the cyclic and corresponding linear polymer. Hence, the peak height should scale as $N^{2\nu_{l}-2\nu_{c}}$, i.e., *N*^0.14^ as is clearly reproduced by data in Figure 5 inset.

Experimental measurements of chain conformation have been largely limited to relatively small molecular weights. However, based on molecular simulations [30,44,91], at very high molecular weight, the rings should follow mass fractal behavior and their dimensions scale with *N*^1/3^. Molecular dynamics simulations carried out by Halverson et al. [34] on rings with up to 1600 monomers per chain indicate that the exponent *ν* varies smoothly with ring dimensions. As the number of monomers per chain increases, the scaling exponent *ν* was predicted to vary from 0.5 (Gaussian chain) to *ν* = 1/3 (crumpled globule) via an intermediate regime with *ν* = 2/5. To fully test this experimentally, large, pure rings are required. Very recently, Iwamato et al. [89] reported a SANS study of the conformation of hydrogenous and deuterated ring PS with *M*_w_ ranging from 10 kg mol^−1^ to 400 kg mol^−1^. Radii of gyration determined from the Guinier approximation were found to scale with the degree of polymerization according to *N*^0.47^, with scaling exponent smaller than expected for Gaussian chains but greater than reported for PDMS and PEO rings, and also expected for crumpled globules. After comparing their results with simulations, the authors concluded that a small amount of impurities due to the presence of linear chains could be responsible for the larger scaling exponents.

## 4. Dynamics of Cyclic Polymers

The dynamic behavior of polymers is important for the understanding of their fundamental physical properties as well as for tuning conditions during processing operations. Despite considerable experimental and theoretical effort in this area, when compared to static properties, polymer motion is still an area of polymer science in continuous development.

One of the motivations in carrying out extensive dynamic studies of rings, at the molecular level, comes from a range of observations based on viscosity and rheological measurements. In 1980, Dodgson, Bannister, and Semlyen reported a study of bulk viscosity of over fifty fractions of cyclic and linear PDMS [12]. The viscosity data of Figure 6 show complex, at that time unexpected behavior: for low *M*, the viscosities for the cyclics, η_c_, are higher than for linear chains, η_l_, but at high *M* the opposite trend is observed [12,13]. Configurational restrictions were believed to be responsible for the higher η_c_ values compared to η_l_, at low *M*. Dynamic differences in this molar mass region are also evidenced by the different slopes of the log η versus log n_n_, which equal to 1.05 ± 0.05 for the linear and 0.60 ± 0.05 for the cyclic fractions. Because of these different dependencies of the viscosity with number of bonds for the two topologies, the ratio η_l_/η_c_ varies with molar mass and only approaches the theoretical value of 2 (based on the ratio of the respective mean square radii of gyration) at ca. 330 skeletal bonds. Above the entanglement molar mass, *M*_c_, η_l_ was found to scale with *M*^3.21^, in line with theoretical predictions. At that time, no theoretical predictions for the viscosity versus molar mass relationship for cyclics had been made. However, it was surprising to find that both the power exponent above *M_c_* (3.46) as well as the *M_c_* value were similar for linear chains and rings. The similarity between power exponents above *M_c_* was attributed to contamination from linear chains at high *M*, a result that was also confirmed by SANS measurements [53], as discussed earlier.

Recently, Pasquino et al. [92] and Richter et al. [44] compared viscosity data at iso-frictional conditions (i.e., same distance from the glass transition) for PS [49,93], polybutadiene [61] and PEO [62]. In all cases, two regimes were identified: (a) for unentangled chains, the ratio between the viscosity of linear chains and rings, η_l_/η_r_ was reported to be constant and equal to 2 but (b) above the entanglement molecular weight and η_l_/η_r_ was found to increase above 2 with increasing number of entanglements. Here we note that, for PEO, η_l_ is nearly twice η_c_ only when comparing rings to the viscosity of linear chains with terminal OH groups [62]. As shown by Nam et al., η_l_/η_r_ equals 1.5 for the polymer with OH end groups but varies from 0.25 to 1 with increasing molar mass of a dimethoxy-terminated linear PEO from 250 to 1500 g mol^−1^ [62]. It was argued that H-bonding between end groups compensates the molecular dependence due to chain-end effects, leading to a ratio that matches the one predicted for chains and rings at theta conditions [62]. While this may be an appropriate argument for PEO, there is evidence that hydroxyl-terminated PDMS may form chain-like or brush-like structures [94] and therefore a direct comparison of its dynamic properties with those of cyclic PDMS would not be meaningful.

Further support to the idea that cyclics are slower compared to linear chains, well below the entanglement molar mass, comes from MD simulations and experimental measurements on PE [95,96]. In their atomistic MD simulations of linear and cyclic alkanes, Alatas et al. showed that cyclic alkanes diffuse more slowly compared to linear chains of equivalent molar mass, a result that was discussed in terms of the higher viscosity of the rings; the presence of rigid and highly symmetric conformers for small rings improves local packing compared to short linear chains [96].

For large rings, all viscosity data available to date (PEO, PDMS, polyisoprene, and PS) indicate that they are faster than linear chains [92]. In the entanglement regime, for PEO, the molar mass dependence of η_c_ was found to be much weaker than that of the linear polymers, more in line with predictions from both scaling theory and simulations compared to data in Figure 6. Specifically, the ratio of the zero shear rate viscosities, η_o,l_/η_o,c_, followed a power-law of type Z^1.2 ± 0.3^ (Z being the number of entanglements), in the range 1 < Z < 20. Polymers other than PEO were also found to obey a similar scaling [92].

Theoretical work and computer simulations have aimed to compare the dynamics of linear and cyclic chains and study the impact of topological constraints with reference to the Rouse and reptation models. The former described the dynamics of short, unentangled chains above *T*_g_, which are considered as a sequence of beads connected by entropy springs of length σ (with σ being of the order of several monomer units) [97].

The main contribution to the understanding of chain motion in a dense solution or melt of entangled linear (or branched) chains is due to the theoretical work of deGennes [98] and Doi and Edwards [99,100,101]. Their reptation model gives a simple physical description of polymer chain dynamics in terms of a snake-like movement within a tube, which accounts for the topological constraints imposed by the surrounding chains. Other theoretical approaches to describe the dynamics of entangled polymer chains have been proposed, but they appear to produce results that are indistinguishable to those obtained by the reptation model [102].

Since the concept of reptation is linked to the existence of chain ends, dynamic studies of non-linear polymer systems for which this motion should be strongly suppressed, e.g., stars and cyclic polymers, can provide an indirect way of testing the model [23,103]. This idea has stimulated a wide range of experimental and computational investigations.

As noted earlier, two neutron scattering techniques (QENS and NSE) can provide molecular level information on the effect of topology on polymer dynamics and therefore offer a tool for comparison with theoretical predictions and computer simulations.

To date, neutron scattering investigations of ring dynamics are very limited [44,54,88,104,105]. In the following sections, NSE and QENS measurements will be reviewed. Although results of NSE studies have been reviewed recently, here our aim is to give an overview of what is possible with neutron scattering, comparing and contrasting these two different neutron probes of polymer dynamics.

### 4.1. Neutron Spin-Echo Studies

Neutron spin echo provides the highest energy or time resolution possible in neutron spectroscopy and can be used to extract information on long-range internal chain motions and center of mass diffusion.

The first NSE measurements comparing cyclic and linear polymers were carried out by Higgins et al. on PDMS in benzene-d_6_ [106]. These authors extracted an inverse correlation time, Ω, corresponding to the first cumulant of coherent intermediate scattering law, *S*(*Q*,*t*):(7)$$\Omega =-\underset{t\to 0}{\lim}\frac{d\;ln\left(S(Q,t\right))}{dt}$$

In the intermediate *Q* region, $R_{g}^{}$ < *Q* < σ^−1^, where σ is the spring length in the Rouse model [97], chain connectivity leads to a universal behavior that is independent of molecular dimensions and molecular structure. At theta- conditions:(8)$$\Omega =\frac{1}{6\pi }\frac{k_{B}T}{\eta_{s}}Q^{3}$$
where *k_B_* is Boltzmann constant, *T* is the temperature, and *η_s_* is the solvent viscosity.

At low *Q*, i.e., *Q·R*_g_ < 1, and times longer than the first Rouse mode, Brownian diffusion of the whole molecules dominates and S(*Q*,*t*) can be simply described by an exponential decay of the form exp(−Ωt). In this regime the temperature-dependent diffusion coefficient can be determined from *D*(T) = Ω/*Q*^2^, in the limit *Q* → 0 [107,108]. Values of Ω/*Q*^2^ for a range of linear and cyclic PDMS fractions are given in Figure 7. It is evident that the NSE measurements provide dynamic information in the cross-over region between diffusion and internal segmental dynamics. Generally, the cross-over point located at *Q*·*R_g_*~1, occurs at lower *Q* compared to the cyclic PDMS with equal M_n_. This arises because, for linear and cyclic PDMS containing the same number of skeletal bonds ${\langle R_{g,l}^{2}\rangle }^{1/2}/{\langle R_{g,c}^{2}\rangle }^{1/2}$ as discussed earlier.

Diffusion coefficients obtained from the horizontal line in Figure 7 were found to be independent of concentration in the experimental range, giving a ratio *D*_l_/*D*_c_ = 0.84 ± 0.016, which is in agreement with the value of 8/3π predicted in the absence of free draining and excluded volume effects [106,109]. Using the Stokes-Einstein equation, hydrodynamic radii, *R_H_*, can be obtained from the diffusion coefficients and the ratio between *R_g_* and *R_H_* compared to theoretical predictions for rings and linear polymers.

No other NSE studies of linear and cyclic polymers in solution seem to have been carried out since this first investigation, with more recent measurements focusing on the dynamics of melts. For these systems, at small angles, NSE is designed to explore length scales close to or above to the entanglement length where reptation dominates.

Computational studies carried out by Halverson et al. on long chains (up to 57 entanglement lengths) indicate that while the Rouse and reptation models satisfactorily describe the dynamics of linear chains, the ring topology leads to a range of different observations, not all easily accounted for [35]. In short, using a semiflexible bead-spring model, these authors reported diffusion coefficients that obey the same power law irrespective of the polymer topology, i.e., *D* ∝ *N*^−2.3^
^± 0.1^. Although rings were found to be faster, meaning that the prefactor was greater than for the linear chains, the molar mass dependence was similar to that expected for entangled linear polymers. However, when considering the zero-shear rate viscosity, η_o_ scaled as *N*^1.4 ± 0.2^ for rings suggesting Rouse-like behavior rather than the *N*^3.4^ dependence characteristic of a reptation mechanism.

For linear chains, the mean-square displacement of a single chain, *msd*, is expected to follow a series of four different power law dependencies in time (Figure 8). At very short times (much shorter than the Rouse relaxation time, τ_R_, Rouse behavior dominates and *msd* ∝ t^0.5^. This is expected to occur until a characteristic time τ_e_ (the Rouse relaxation time at the entanglement length N_e_) is reached. When the *rms* displacement of the segments becomes comparable to the tube diameter, *msd* ∝ t^0.25^; the chain segment is restricted in the direction tangent to the primitive path of the tube follows Rouse motion within the reptation tube. At longer times, for t > τ_R_ the chain diffuses along the tube and a second t^0.5^ dependence is observed. Finally, for times longer than the reptation time, τ_rep_ free diffusion is observed with a t^1^ dependence.

Interestingly, for rings, Halverson et al. observed a similar slowing down from t^0.5^ to t^0.25^, which in this case, cannot be simply interpreted in terms of reptation, and therefore the reasons for observing this time regime in cyclic polymers are not clear [35].

The first NSE investigation of melt dynamics of rings was carried out on PEO rings by Bra et al. [54,88]. The coherent intermediate scattering function measured by NSE for cyclic and linear PEO (*M*_w_ ca. 5000 g mol^−1^) at the lowest experimental *Q* value (=0.05 Å^−1^) is shown in Figure 9 [88]. One should note that even at this low *M*_w_ value, the PEO sample is above the entanglement molecular weight (Table 1).

The less pronounced decay of *S*(*Q*,*t*)/*S*(*Q*,0) for linear PEO (Figure 9), confirms that linear chains are slower compared to rings of similar molar mass. Only the first part of the Rouse equation, i.e., $S\left(Q,t\right)\propto exp\left(-Q^{2}Dt\right)$ was required to fit data in Figure 9, thus giving a measure of the diffusion coefficient of the center of mass, *D*. Values of *D* were reported to vary from 2.05 ± 0.02 Å^2^ ns^−1^ for rings to 0.94 ± 0.01 Å^2^ ns^−1^ for the linear chains. This leads to a ratio between diffusion coefficients equal to 2. The same ratio was measured for linear and ring polymers with degree of polymerization of ca. 40 [54]. Deviations from the fits seen for the linear chains at short times in Figure 9 were attributed to entanglement effects, which are active in linear matrices but absent in melts of rings.

It has been noted that because *D* in the Rouse model depends only on the segmental friction coefficient and the degree of polymerization, it should be independent of chain topology [54]. However, differences between *D* values are in line with a range of experimental observations such as difference between the radii of gyration, hydrodynamic radii, as well as the ratio between the viscosity of linear chains and rings [110].

Due to their symmetry, for rings, only even internal Rouse modes contribute to *S*(*Q*,*t*) and this made it possible to determine the center of mass mean square displacement, < r^2^_cm_(t) >, from the *Q* dependence of *S*(*Q*,*t*)/*S*(*Q*,0) data. [44,111] For PEO rings with *M*_w_ varying from 5000 to 20,000 g mol^−1^, just below the cross-over to the diffusion regime when < r^2^_cm_(t) > ∝ t^1^, < r^2^_cm_(t) > was found to follow a sub-diffusive behavior of type < r^2^_cm_(t) > ∝ t^3/4^ within a region that extends in space up to several times the *R_g_* of the rings. This finding is in agreement with simulations carried out by Halverson et al. who also predicted a sub-diffusive < r^2^_cm_(t) > behavior with t^3/4^ dependence. [35] Departures from this behavior and a weaker time dependence are observed with increasing, at short times, which appears to be consistent with Monte Carlo simulations of Milner and Newhall who developed a theory for stress relaxation of unconcatenated rings based on “lattice animal” configurations [112].

The time dependence of the mean square displacement < r^2^(t) > for a PEO ring sample with *M*_w_ equal to 20,000 g mol^−1^ is given in Figure 10. As discussed above, for low molar mass polymers at low *Q*, the main contribution to the NSE data is from the center of mass diffusion. However, at higher *Q* values and at higher *M*, internal segmental motion will also contribute and therefore a more detailed Rouse analysis needs to performed. This may be challenging, as shown by the work of Gooßen et al., who were able to demonstrate that PEO ring sections of about 60 monomers relax undisturbed by topological effects at short times, a result that was attributed to free Rouse relaxation of loops formed within a lattice animal structure. [44,111] These loops have a molecular weight of about 2500 g mol^−1^, which are of the order of the entanglement length for PEO (2000 g mol^−1^). At longer times, slower loop motion is observed with a t^0.32^ dependence.

### 4.2. Quasielastic Neutron Scattering

The QENS technique can be used to observe polymer dynamics on nanosecond to picosecond time scale and length scales up to a few statistical segment lengths. Therefore, at temperatures above the glass transition, the technique provides access to the internal relaxation processes of a polymer including Rouse modes, segmental motion of the chain, and rotation of side groups, if present. Due to the limited time and distance scale of observation, motion detected by QENS is largely determined by intra-molecular potentials and well outside the range required to explore the effect of chain entanglements on chain motion.

QENS has been extensively used over many years to study local dynamics in polymers. However, there is little in the literature on the comparison between cyclics and linear chains, except for QENS measurements on PDMS [104,105,113]. The local dynamics of linear PDMS has been investigated extensively by neutron scattering [114,115,116], dielectric spectroscopy [117,118,119,120], NMR [17], as well as simulations [121], and these studies provide a starting point for a detailed comparison with ring dynamics.

In 1972, Allen et al. [104,113] carried out QENS experiments on linear and cyclic PDMS samples with degree of polymerization less than 20 (equivalent to *M*_n_ ≤ 1500 g mol^−1^). Effective diffusion coefficients were extracted from the quasi-elastic broadening and these are plotted in Figure 10 as a function of the number of silicon atoms (equal to the number of repeat units). Contrary to the NSE measurements of PDMS in benzene-d_6_ discussed earlier, here the short linear PDMS chains are seen to be faster than the small rings, the difference becoming less pronounced with increasing molecular weight (Figure 11). This finding that cyclic PDMS is slower compared to linear chains having the same number of chemical bonds, although apparently at odd with recent literature [44,54,88], is consistent with the viscosity measurements due to the very low molecular weight of the samples coupled with the fact that QENS probes dynamics at length scales of the order of a few statistical units.

In a recent study, we carried out similar QENS measurements on a series of linear and cyclic PDMS fractions over a wide range of molar mass and at approximately 110 degrees above their *T_g_*s [105]. As for previous QENS analysis of high molecular weight linear PDMS [114,115], the dynamic incoherent structure factor was described by the convolution of two functions, one representing the local segmental relaxation, and the other one the rotational motion of the CH_3_ groups, which, as shown in Figure 12, make a non-negligible contribution to the quasielastic broadening.

The narrower component in Figure 12 was modeled by the Fourier integral of the KWW function [114,115] describing segmental motion [122,123]. From this, two parameters can be obtained: (a) the stretched exponent β, which characterizes the width of the distribution of relaxation times and (b) the characteristic time τ of the molecular motion. To account for differences in the distribution of relaxation times among data sets, the effective time τ*_eff_* can be calculated from:(9)τeff=τΓ(1β)β

Only small differences in the stretched exponent β were observed, but values were reported to decrease with increasing molar mass for both linear chains and cyclics, approaching a constant value at high *M* equal to 0.56 and 0.52 for linear and cyclic PDMS, respectively. This followed the trend predicted by theoretical calculations of Ganazzoli et al., applicable to both simple chain models and real chains [124]. The same methodology was used to compare experimental QENS data of linear and cyclic PDMS with theoretical predictions based on a realistic representation of a PDMS chain within the Rotational Isomerical State approach [105].

The few studies of local chain dynamics that have been published give contradictory results. For example, Krist et al. reported no change in the distribution of relaxation times with topology, *β* being equal to 0.485 for both linear chains and rings [117]. Goodwin et al. found a similar *β* average value, i.e., 0.48, for linear PDMS but a greater one for rings (*β* = 0.53) [120], suggesting a more cooperative relaxation in linear PDMS.

As shown in Figure 13, rings have longer relaxation times compared to linear chains of the same number of monomer units, a result that is also supported by theory [105]. Furthermore, relaxation times of both linear and cyclic PDMS increase with increasing molar mass. According to theoretical calculations, the observed slowing down of local dynamics is due to the topological constraint imposed by the ring closure, which becomes negligible for very large molar masses. Calculations suggest that, due to its albeit small conformational rigidity, cyclic PDMS undergoes an additional constraint, which further increases the relaxation time, producing a shallow maximum for N ≈ 50 repeat units. A similar feature can be seen in the experimental QENS data of Figure 13 (broad maximum in τ*_eff_*).

The experimental QENS data [104,105,113] and theoretical calculations [105] are supported by bulk viscosity [12,13,14], self-diffusion, and spin-spin relaxation measurements [17], according to which, rings have slower dynamics (higher viscosities or smaller diffusion coefficients) compared to linear chains at low molar mass. As noted by Semlyen et al., configurational restrictions in rings are likely responsible for a slowing down of the segmental mobility, resulting in higher viscosity at low molecular weight [12,13,14].

To date, only PDMS has been studied by QENS and so these results cannot be compared to other cyclic systems. However, MD simulations carried out by Tsolou et al. [110] show that the segmental dynamics of small polyethylene rings is slightly slower than that of linear chains, the difference decreasing as M increases. This effect was attributed to the extra stiffness of small rings. Similarly, self-diffusion, NMR spin-spin relaxation, and zero shear rate viscosity measurements of low molecular weight cyclic (400 to 1500 g mol^−1^) and linear PEO samples carried out by Nam et al. [62] show slower dynamics for rings. As noted earlier and shown by Nam et al., when hydroxy-terminated rather than dimethoxy-terminated linear PEO samples are compared to the PEO rings, the opposite trend is observed, the cyclics being faster than the linear chains. This effect is due to the additional strong interactions between the OH end groups.

## 5. Blends of Polymers with Different Topologies

Mixing of low molar mass species is dominated by the configurational (or ideal) entropy of mixing, but this contribution to the free energy of mixing diminishes rapidly as molar mass increases so that mixing of polymeric molecules is highly sensitive to a range of parameters beyond temperature and concentration such as specific intermolecular interactions, chain architecture, and tacticity [125,126]. Binary mixtures of linear and cyclic polymers provide a simple model system to study the influence of topology on blend thermodynamics.

Contamination from linear chains has been considered the main cause for many of the anomalous experimental results on cyclic materials [40]. For example, as reported by Kapnistos et al. [40], a small amount of linear chains in a melt of rings can greatly affect their rheological behavior. Therefore, understanding the dynamics of the individual components in blends of the two different topologies is useful not only in order to clarify a range of experimental findings reported in the literature but also to better understand molecular motion of entangled rings and linear polymers [81,127].

### 5.1. Miscibility and Conformation

Understanding topological effects in miscibility of blends is therefore an important area [77,128], but only limited studies have been performed. As suggested by Cates and Deutsch [22] and Khokhlov and Nechaev [77], enhanced compatibility should be observed when mixing rings with linear chains compared to blends of linear chains.

In 1986, Cates and Deutsch [22] predicted that blends of chemically identical linear chains and rings should display a negative Flory-Huggins interaction parameter. As pointed out by Sakaue and coworkers [128,129], the relaxing of the entropic penalty due to non-concatenation in melts of rings is the driving force towards mixing of linear and cyclic polymers. The authors introduced the idea of *topological volume* according to which the topological constraints are represented by an effective excluded-volume with the topological length playing the role of a screening factor [128,130,131]. The linear chains act as diluents, softening the topological restrictions imposed on rings and this results in a topological entropic gain for blends of linear and cyclic polymers.

The experimental findings of, for example, Nachlis et al. [132] on cyclic bisphenol A carbonate oligomers in linear polystyrene melts, Santore et al. [133] on cyclic polystyrene/linear poly(vinyl methyl ether) (PVME) blends, as well as Kuo and Clarson [134] on oligomeric cyclic methyl phenyl siloxanes in siloxane melts support these predictions. As noted by Singla and Beckham [135], replacing a linear component with the cyclic analogue may provide a route to improve blend miscibility among immiscible linear polymers. This was also confirmed by their ^1^H solid-state NMR and thermal measurements on blends of cyclic poly(oxyethylene) and PS.

Although a range of experimental techniques can be used to assess whether topology affects miscibility, SANS measurements can provide information otherwise unattainable. For example, using mixtures of hydrogenated cyclics in melts of deuterated linear chains, it is possible to test theoretical predictions that cyclic polymers are expanded in the presence of long linear chains [22].

Very few SANS measurements have been reported. Gooßen et al. compared the conformational properties of a PEO cyclic sample (M_n_ = 20,100 g mol^−1^) in a deuterated linear matrix with that in a melt of rings and reported a 20% increase in the radius of gyration for the topological blend [44,127]. This appears in line with both theoretical and computational studies indicating that, when linear chains are added to rings in bulk, the former thread into the rings, and the latter consequently swell [79,80,81]. Iyer et al. [80] noted that, when gradually substituting some of the ring polymers with linear chains, the ring molecules swell. In the limit of infinite dilution, their size scales as *R_g_* ∝ *N*^0.5^.

MD simulations of PEO rings by Tsalikis and Mavrantzas [136] showed that radii of gyration of rings are considerably larger when these are mixed with linear chains compared to when surrounded by an environment of similar rings. The greatest degree of swelling was found for large rings in short linear polymers, a result that was attributed to excess free-volume effects due to chain ends, with the lower density of small chains favoring more open ring structures. As for the linear chains, it was found that short ones were unaffected by the presence of rings but larger ones expand.

A more extensive SANS study has been carried by Kobayashi et al. [137] on blends of poly(4-trimethylsilylstyrene) (PT) and polyisoprene (PI) with different topology (linear-linear, ring-linear, and ring-ring blends). In each of these blends, one component is deuterated. All three blend systems were found to exhibit a lower critical solution temperature (LCST) behavior.

At low *Q*, the structure factor *S*(*Q*) obtained from the SANS scattered intensities was modelled according to the Ornstein−Zernike equation [41]:(10)$$\frac{1}{S\left(Q\right)}=\frac{1+Q^{2}\xi^{2}}{S\left(0\right)}$$
where ξ is the correlation length of the concentration fluctuations in the blends, and *S*(0) is the extrapolated value of the structure factor at *Q* = 0. The latter can be expressed as:(11)$$\frac{1}{S\left(0\right)}=2A\left(\frac{1}{T_{s}}-\frac{1}{T}\right)$$
where *A* is an enthalpic term and *T_s_* is the spinodal point. The temperature dependence of the ξ^−2^ and 1/*S*(0) are shown in Figure 14, together with extrapolations based on Equation (10).

As shown in Figure 14, the spinodal temperature for the ring-ring blend is much lower compared to that of the other two systems. Although this indicates that miscibility is considerably suppressed in R-R blends, as expected, the similarity between *T_s_* values for the L-L and L-R mixtures seems to contradict theoretical predictions that the mixing should be more favorable in the L-R system [128]. However there is a rather long extrapolation in temperature to obtain the spinodal, and a small adjustment to the slope of the L-R data could produce a difference in *T_s_* of up to 30 °C. Moreover, since only one concentration was shown for each blend, any shift in the shape of the phase boundary cannot be detected. Optical microscopy (OM) experiments on the same system but using only non-deuterated samples also showed LCST behavior, but the phase boundaries were some 100 °C below the spinodals for the deuterated samples. This deuteration effect in polymer blends is well known and another example of how small differences can produce large changes in the thermodynamics of mixing [138]. Lipson et al. [138] have shown that deuteration changes the thermal expansion coefficient of a polymer, and an increased similarity of the expansion coefficients of the blend components can introduce large changes in the miscibility limits. The OM data in reference [137] show the whole phase diagram for the L-R and L-L blends and it can be seen that there is a small shift in *T_s_* for the L-R blend, but also an important shift in the critical composition.

Further light is thrown on the thermodynamics of R-L systems by some very interesting observations made by Garas and Kosmas [139] in their theoretical study of R-R, R-L and L-L blends, according to which R-R considerably deviate from the Flory—Huggins theory and have highly concentration-dependent χ parameters, as well as χ_eff_ values as determined by SANS. Both ring and linear chains are expected to contract when the binary system approaches the phase separation limit. These predictions could be, in principle, tested by SANS experiments.

However, as noted by Kobayashi et al., the analysis of the *Q* dependence at higher *Q* values in order to explore the chain conformation in the R-L blend is severely hampered by the lack of appropriate expressions for *P*(*Q*) to use in the RPA analysis as in Equation (2) [137].

### 5.2. Dynamics

Addition of linear chains to cyclic polymers is known to greatly affect diffusion [15] as well as rheological behavior, as shown by Roovers [61], McKenna and Plazek [140,141]. In their study of dilute PS rings in linear matrices, Mills et al. observed a pronounced slowing down of ring diffusion with increasing molar mass of either rings or linear chains, in symmetric blends [15]. Their results were discussed in terms of three diffusive mechanisms: (a) reptation of unthreaded rings in the confining tubes defined by the matrix of linear chains, as first proposed Klein [23], (2) constrained diffusion along the contour length of the host chains by those rings threaded only once by long linear chains, and (3) diffusion of multiply threaded rings via constraint release of the threading host chains. For low molecular weight ring/linear PDMS blends, Cosgrove et al. [18] also invoked a ring-threading mechanism to account for the dynamics, above a critical ring size of approximately 33 monomers.

Considerable computational work has been carried out to understand the effect that linear chains have on ring dynamics [32,36,79,96,142]. In agreement with pulsed-gradient spin-echo nuclear magnetic resonance data of von Meerwall et al. [95] on cyclic and linear alkanes, the MD simulations of Alatas et al. [96] showed that rings have higher densities and diffuse more slowly than the equivalent linear alkanes, a result that was explained in terms of the higher density of the small cyclic molecules at the same temperature. The overall diffusion coefficient of linear/cyclic blends having the same number of carbon atoms was found to be equal to the weight-average value of the self-diffusion coefficients of the pure components, indicating that a simple concentration dependent viscosity and local dynamic cooperation between the two components [96].

MC simulations of Subramanian and Shanbhag [142] of relatively large polymers suggested that rings diffuse much slower in blends with linear chains of similar molar mass compared to their own melt. This effect was seen to be more pronounced with increasing molar mass. By contrast, the dynamics of the linear chains remained unaffected, a result that is supported by coarse-grained MD simulations of Halverson and coworkers [79]. Interestingly, for asymmetric blends where chains have smaller *M*_w_ compared to the rings, ring diffusion showed a maximum at intermediate *M*_w_ values for host chains. This was attributed to the increased number of threading events with decreasing *M*_w_ of the linear chains.

A very detailed, systematic MD study of linear/cyclic PEO blends was carried out recently by Tsalikis et al. who showed that the *M*_w_ dependence of the ring diffusion coefficient follows a power law of type *D* ≈ *M*_w_^−*b*^ with *b* increasing with increasing length of the linear chains [36]. Blends of rings in short chains showed Rouse scaling, in agreement with PFG NMR measurements of Kruteva et al. [143]. For very short linear matrices, due to chain-end free volume effects, a weaker dependence on chain length (*b* < 1) was observed. More interestingly, power law exponents well above 1 were obtained with increasing *M*_w_ of the linear host chains. Detailed analysis of the MD trajectories led to the conclusion that, irrespective of the ring or linear chain molecular weight, all rings were threaded by at least one chain. In principle, both ring-ring and ring-linear chain threading could occur. However, the occurrence of ring-ring threading was negligible, ring-linear chain threading being the main mechanism affecting the segmental and diffusive dynamics of the rings but also their size and conformation [36].

To date, neutron studies of linear/cyclic blend dynamics are limited to a few NSE measurements on PEO; no QENS data have been reported. Nevertheless, in their NSE study of topological blends, Gooßen et al. [127] showed that the segmental dynamics of PEO rings is an extremely sensitive probe of the environment generated by the host molecules, i.e., the linear PEO chains. In particular, the intermediate scattering function, *S(Q*,*t)* of large PEO rings (*M*_w_ = 20,000 g mol^−1^) in highly entangled linear matrices (*M*_w_ = 80,000 g mol^−1^) displays very different dynamic behavior compared to the same rings mixed with very short chains (*M*_w_ = 2000 g mol^−1^). As shown in Figure 15, after an initial decay, *S*(*Q*,*t*) reaches a plateau that is *Q* dependent. Furthermore, the dynamics of rings mixed with linear chains is considerably different from that of a melt of pure rings (see inset of Figure 15) both at long and at short times. As noted by Gooßen et al. [127], there is no sign of the loop dynamics or of the loop migration observed for the PEO pure rings. The more pronounced decay at short times suggests more freedom of movement in the blend. After ca. 20 to 40 ns, *S*(*Q*,*t*) reaches a plateau from which a confinement size equal to *d_e_* = 42 Å was obtained (the tube diameter for PEO linear chains is reported to be *d* = 52.5 Å [144]). Since the rings cannot undergo local reptation as linear chains do, they probe the constrains imposed by the linear chains, unaffected by the relaxation of longitudinal Rouse modes. This is reflected by the different values, *d* and *d_e_* for the relaxed and unrelaxed constraints.

The NSE measurements of Gooßen et al. [127] find support in the atomistic MD simulations of Papadopoulos et al. [32] of large PEO rings in linear chain matrices of varying molecular weight. Good agreement between calculated and experimental NSE spectra was reported, and between diffusion coefficients [32]. A detailed geometric analysis of the MD simulations indicated that significant threading by linear chains occurs. Although the small concentration of rings did not appear to affect the average size of the tube, the dynamics of the rings at time longer than τ_e_ is determined by ring-linear disentanglement events. This is responsible for the *S*(*Q*,*t*) plateau observed at long times in MD simulations and experiments.

## 6. Summary: Experimentally Confirmed Theory and Open Questions

### 6.1. Conformation, Scaling, and Relative Sizes of Rings and Chains

SANS from dilute solution of cyclics and linear analogues confirms theoretical predictions as well as simulations for molecular size both in terms of the overall dimensions (*R_g_*), the ratio of *R_g_* values for equivalent *M*_w_ linear and cyclics, and the scaling of *R_g_* with molecular weight (*ν* exponent) for solvents of varying quality, including theta solvents. Any disagreements seem to be reasonably explained by the contamination of the cyclics by small linear fractions.

Because of the difficulty of synthesizing deuterated cyclics, experiments on bulk samples have been limited to relatively few polymers, specifically PDMS, PEO, and PS and molecular weights typically up to 20,000 g mol^−1^, with only few SANS measurements on higher molecular weight PS rings [74,89] as discussed earlier. Results for *R_g_* and *ν* agree with predictions that the cyclics are partially collapsed in the melt with a value *ν* of 0.43. However, the predicted collapse into globules for high molecular weight rings (which has been seen in simulations) has not yet been observed [89]. The particle form factor calculated by Casassa fits the low *Q* data at least for the smaller molecular weights, and the predicted peak in the Kratky plot is nicely observable at *Q*·*R_g_* around 2. However, there is still a lack of better models for the particle form factor of rings in bulk samples covering the whole *Q* range. The unified model introduced by Beaucage et al. has some success in fitting the data by stitching together the Debye model for low *Q* with a fractal model at high *Q*. As discussed by Hammouda [145], the Beaucage model is particularly useful when modelling form factors that involve numerical integrations as it is the case for cyclic polymers [90]. Although it undoubtedly speeds up analysis of SANS data, it appears to work best for mass fractals with Porod exponents between 5/3 and 3. We note that the expression for the form factor for a monodisperse ring of arbitrary molecular weight has been reported by Hammouda [146] and used by Bra et al. [54] in their structural study of PEO rings. This has also been extended to rings with excluded volume interactions [90].

In binary, topological mixtures, theory and simulations predict that the linear molecules can thread through the rings allowing swelling of the ring molecules. In blends, replacing one of the linear components with the cyclic analogue may somewhat increase miscibility, but when both components are cyclic molecules it is severely reduced. Both these predictions have been confirmed for a number of blends by SANS and other experiments.

### 6.2. Local and Large Scale Dynamics

Understanding the relaxation mechanisms of melts of rings is very challenging and requires high molecular weight cyclics, free from contamination [44,92,147]. Despite many controversial observations (largely due to the difficulty of obtaining 100% pure cyclic polymers), experiments have shown that moderately large rings with molar masses above the entangled regime do not exhibit an entanglement plateau, unlike linear chains [148]. Based on rheological data [12,13,14] and measurements of diffusion coefficients [15,16,17,18], rings have been found to diffuse faster than the corresponding linear chains. Interdiffusion experiments of bilayers consisting of cyclic PS and deuterated cyclic PS have also shown faster mutual diffusion of rings compared to equivalent bilayers of linear chains [149]. These observations are supported by many computer simulations predicting not only faster diffusion but also shorter relaxation times for melts of rings compared to linear chains of equal mass [24,25,26,27,28,29,30,31,35].

For PDMS, in solution, *D*_l_/*D*_c_ was found to be equal to 0.84, in agreement with the value predicted in the absence of free draining and excluded volume effects [106]. As expected from their smaller size, rings diffuse faster compared to linear chains. The opposite behavior was observed in QENS measurements of local dynamics, for relatively small PDMS rings [104,105]. This is not surprising in view of the viscosity data of Semlyen et al. (see Figure 6), which showed a cross-over from slower to faster dynamics with increasing molar mass. As discussed in previous sections, as the molar mass of the rings and chains decreases, due to chain end effects (for linear polymers) and topological restrictions (for rings) the relative dynamic response changes, with cyclics becoming slower than linear chains. Similar trends have been reported by Ozisik et al. [150], Hur et al. [27,28], and Alatas et al. [96] in their computer simulations of cyclic and linear polyethylenes. It is, however, still unclear whether the cross-over from slower to faster ring dynamics is a common feature to all cyclic systems. As shown by work on linear PEO [62] and linear PDMS [94], the presence of strongly interacting end groups alters the viscosity and other physical properties. In this respect, at small *M*, the choice of linear polymer to be compared with the cyclic molecules will strongly determine the outcome, e.g., the ratio *η*_l_/*η_c_* and conclusion regarding the relative dynamic behavior.

Although very limited neutron studies have been carried out to date, the conclusions that can be drawn by the NSE experiments on PEO [44,54,88,111] as well as the NSE and QENS measurements on PDMS [104,105,106] give a molecular picture of ring dynamics that is consistent with a range of experimental measurements, theory, and MD simulations.

Local chain dynamics as measured by QENS are unaffected by the presence of entanglements: characteristic times approach constant values at high *M*_w_ that appear to be similar for rings and linear chains [105]. This is not the case for NSE measurements carried out on linear and cyclic PEO at longer times and larger distances [44,54,88,111].

In the NSE regime, not only rings are faster than linear chains of similar molar mass, but they also display unique features, not seen for entangled linear polymers. These include, for example, absence of a plateau in the coherent intermediate scattering function at long times, which is a characteristic feature of entangled chains and a signature of reptation. A sub-diffusive regime with <r^2^_cm_(t)> ∝ t^3/4^ was identified, in agreement with simulations carried out by Halverson et al. [35]. Interestingly, the NSE data provide evidence of a basic length scale, i.e., the loop size, of the ring molecule and loop motions [44].

It is when rings and chains are observed in mixtures that a whole range of new behavior is seen to arise, and where there is a need for much more experimental work. NSE experiments of dilute solutions of PEO rings in linear melts clearly indicated that these are model systems for understanding the dynamics of the entangled linear chains [127].

## Figures and Tables

**Figure 1 polymers-12-01884-f001:**
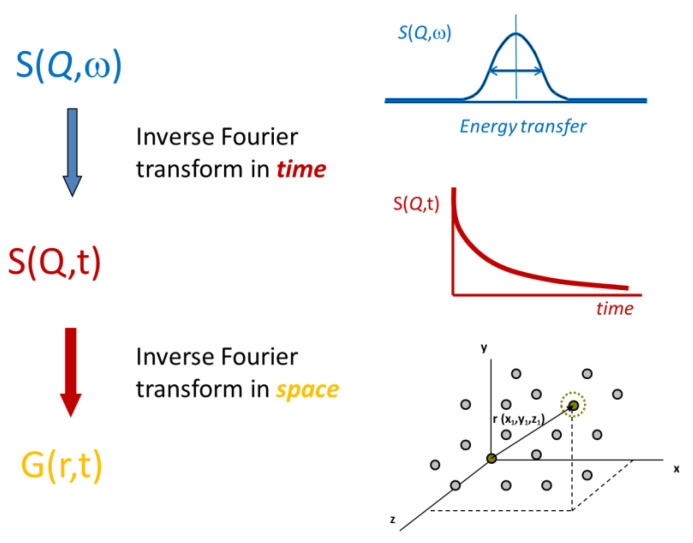
Schematic representation of the information obtained from QENS and NSE experiments. QENS probes molecular motion in the frequency domain through the dynamic structure factor, *S*(*Q*,*ω*). The QENS signal is the Fourier transform of the time correlation function, *S*(*Q*,*t*), which is directly measured in NSE experiments. Inverse Fourier transform in space yields either the pair or the self-correlation function here generally indicated as *G*(*r*,*t*).

**Figure 2 polymers-12-01884-f002:**
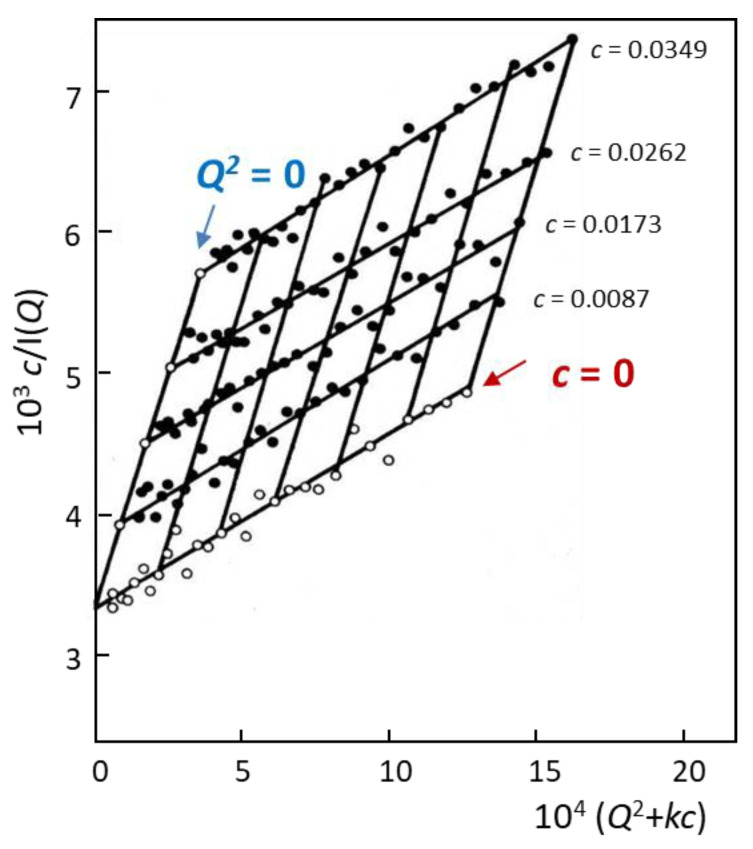
Zimm plot of the scattered intensity *I*(*Q*) (cm^−1^) of solutions of cyclic PDMS (z-average molar mass, *M*_z_ = 20,210 g mol^−1^) at four different concentrations *c* expressed in g cm^−3^. Closed circles represent experimental data and the open circles are points obtained through extrapolation at zero concentration and *Q* (Å^−1^) equal zero. All samples are in benzene-d_6_, measured at 292 K. Adapted from reference [46].

**Figure 3 polymers-12-01884-f003:**
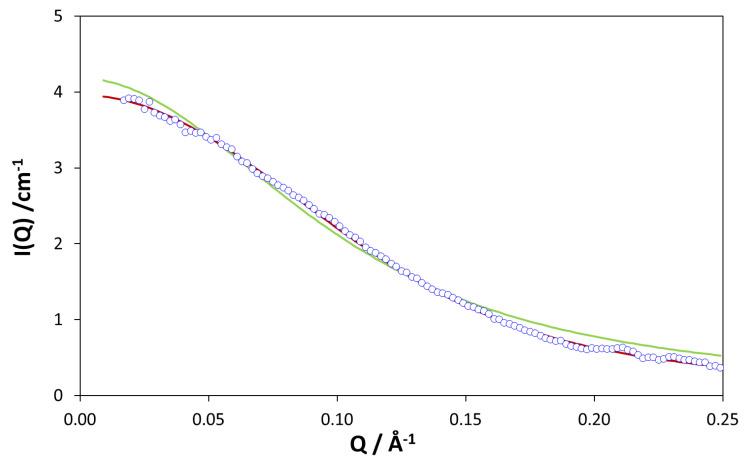
Experimental data and best fit as obtained for a blend of hydrogenated (*M*_w_ = 4800 ± 500 g mol^−1^) and deuterated cyclic PDMS (*M*_w_ = 4800 ± 500 g mol^−1^) at *Φ_H_* = 0.52 using either the Debye (green line) or the Casassa (red line) form factor. All fits were performed by letting both interaction parameter and radius of gyration vary. Adapted from reference [53].

**Figure 4 polymers-12-01884-f004:**
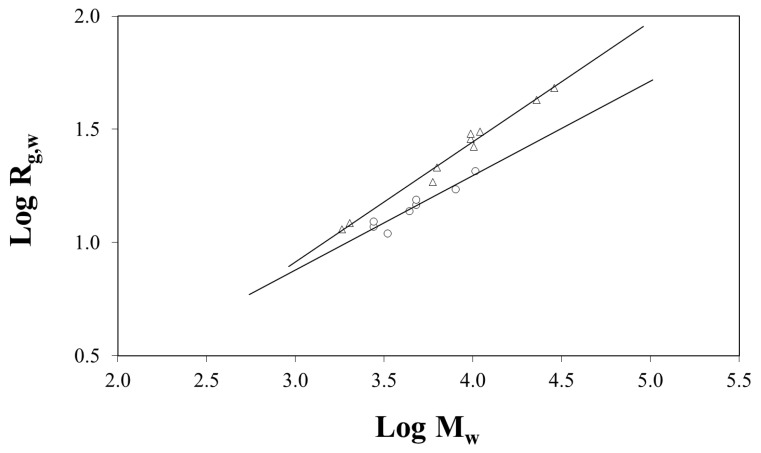
Weight-average radii of gyration vs. *M*_w_ for the linear (Δ) and cyclic (O) H/D PDMS blends. Adapted from reference [52].

**Figure 5 polymers-12-01884-f005:**
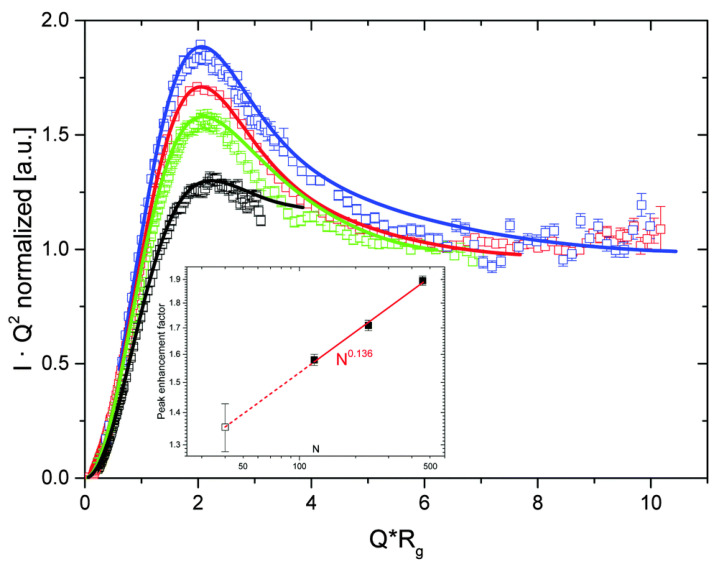
Kratky plot (scaled with *R_g_*) of SANS data from PEO rings with molecular weights 2 k (black symbols), 5 k (green symbols), 10 k (red symbols), and 20 k (blue symbols). Adapted from reference [44]. Inset: peak height as a function of *N*. Reproduced with permission from reference [44].

**Figure 6 polymers-12-01884-f006:**
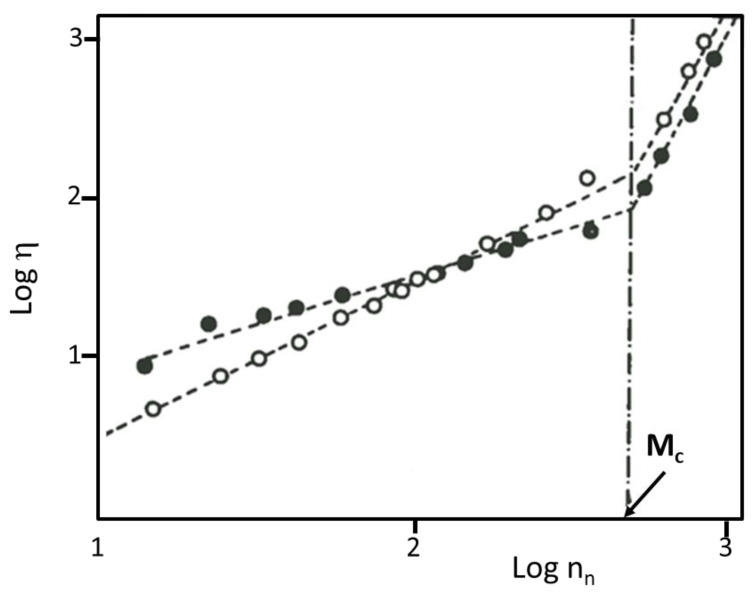
Molar mass dependence of bulk viscosity for ring (O) and chain (•) PDMS fractions. To compensate for changes in segmental mobility viscosity data acquired at 298 K were scaled at constant *T*-*T*_g_. Data are plotted versus the number average number of skeletal bonds, n_n_. Adapted from reference [13].

**Figure 7 polymers-12-01884-f007:**
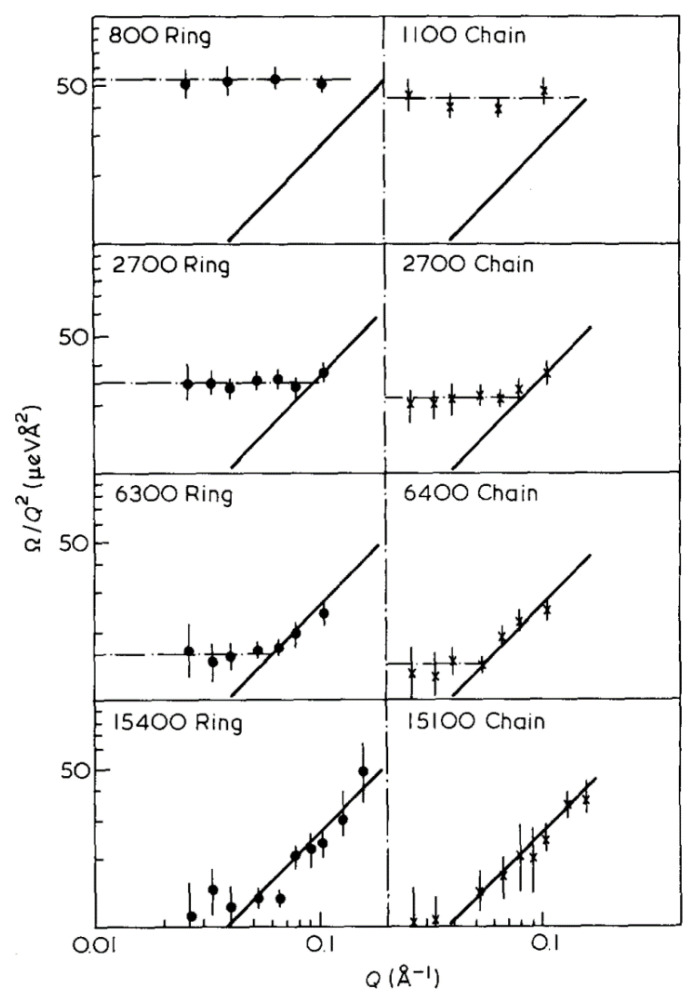
Experimentally determined behavior of Ω/*Q*^2^ for ring (R) and linear chain (L) PDMS fractions of different average molar mass, as indicated, in solution in benzene-_d6_. The broken lines are the experimental) values of *D*(c) and the solid lines represent the *Q* dependence predicted by Equation (7). Reproduced with permission from reference [106].

**Figure 8 polymers-12-01884-f008:**
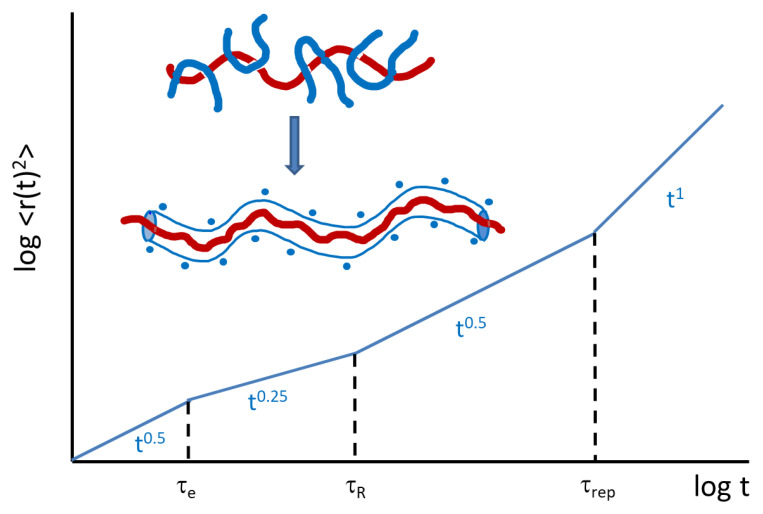
Dynamic properties of entangled polymer chains, according to the reputation model.

**Figure 9 polymers-12-01884-f009:**
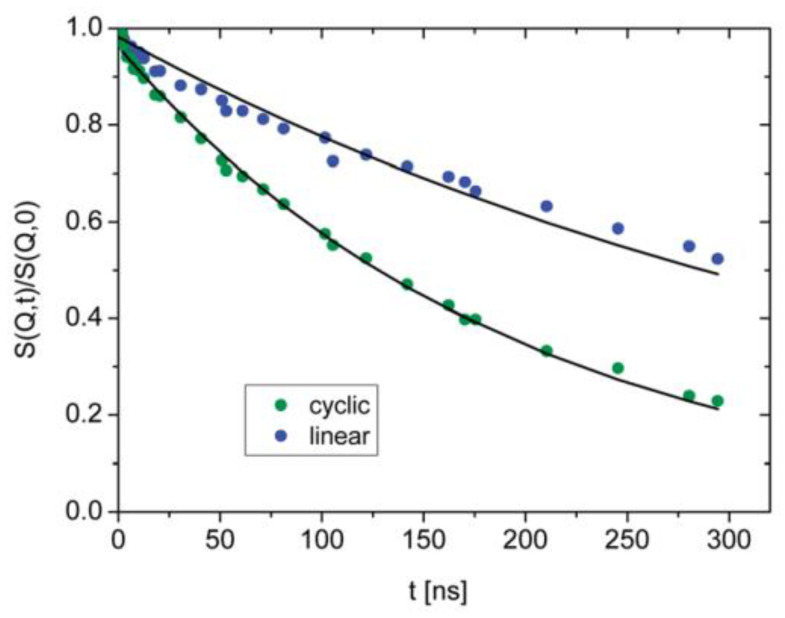
Intermediate scattering function, *S*(*Q*,*t*)/*S*(*Q*,0), for a PEO ring and a linear chain of *M*_w_ equal to ca. 5000 g mol^−1^, at the lowest experimental *Q* value (0.05 Å^−1^). Reproduced with permission from reference [88].

**Figure 10 polymers-12-01884-f010:**
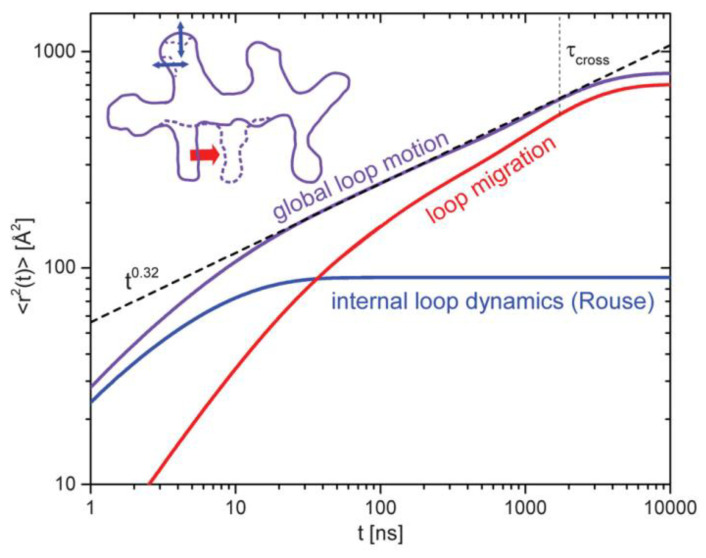
Time dependence of the mean square displacement of a PEO ring sample with *M*_w_ = 20,000 g mol^−1^. Loop relaxation occurs at short times (blue line), whereas the dynamics of the loops is visible at longer times (red line). The purple line is the sum of the two contributions. A t^0.32^ dependence is seen at intermediate times (dashed black line). Reproduced with permission from reference [111].

**Figure 11 polymers-12-01884-f011:**
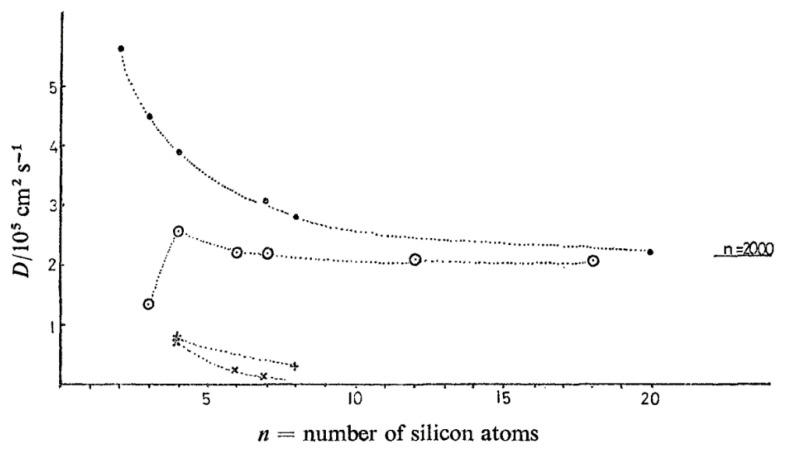
Effective diffusion coefficient, *D_,_* for ring (O) and linear PDMS (•). Spin-echo NMR data for rings (×) and PDMS chains (+) are also reported. All measurements were carried out at room temperature. Reproduced with permission from reference [104].

**Figure 12 polymers-12-01884-f012:**
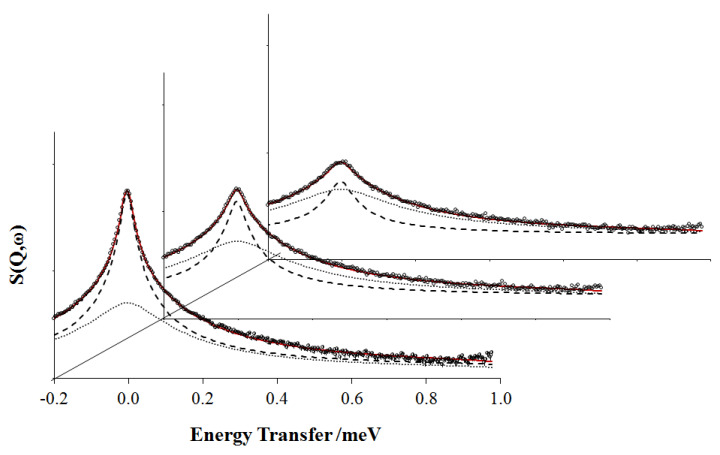
QENS spectra of cyclic PDMS (*M*_w_ = 2700 g mol^−1^) at 283 K and *Q* = 0.92, 1.42, and 1.76 Å^−1^ (from front to back). Symbols represent experimental data. Full lines are fits using Equation (24) in reference [105]. The dashed and dotted lines represent the two dynamic contributions: (a) segmental dynamics (dashed) and methyl rotations (dotted). For clarity, error bars are only shown for one set of data (*Q* = 0.92 Å^−1^). Reproduced with permission from reference [105].

**Figure 13 polymers-12-01884-f013:**
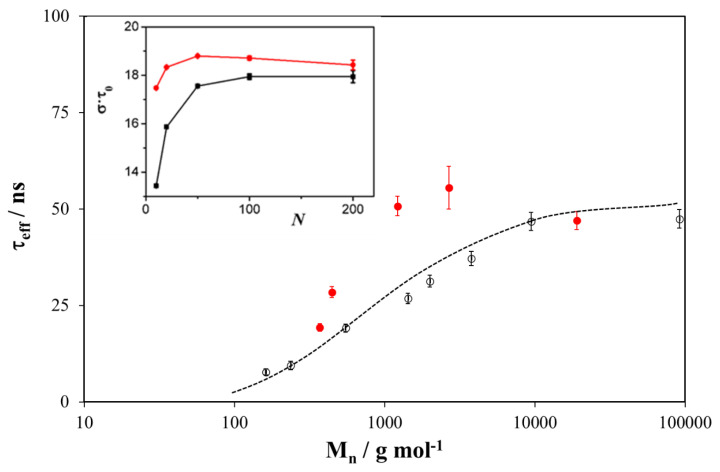
Molar mass dependence of τ*_eff_* for linear (O) and cyclic (•) PDMS at ca. 110 degrees above the corresponding glass transition. The dotted line is a guide to the eye. Error bars are shown for all data. Inset: effective characteristic time *τ*_0_ (σ is the time unit, see Equation (11) in reference [105]) plotted as a function of N for the PDMS chain model. Adapted with permission from reference [105].

**Figure 14 polymers-12-01884-f014:**
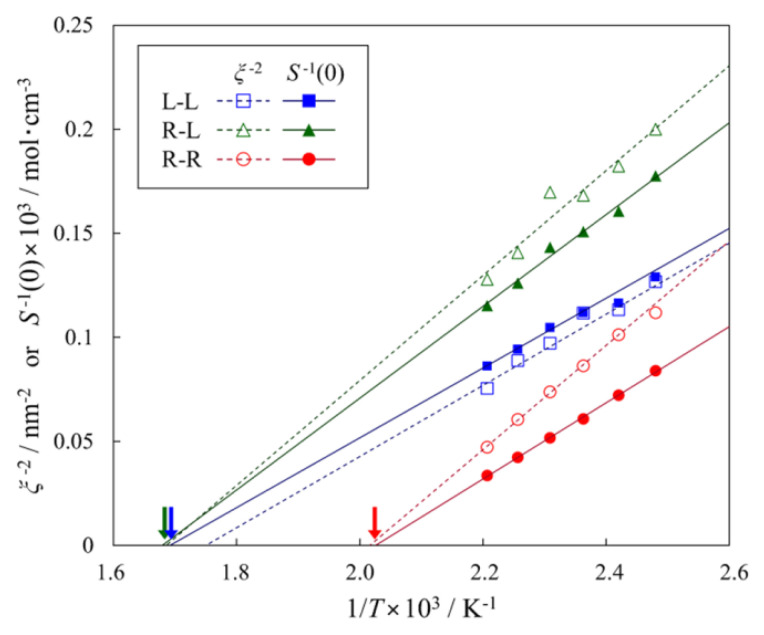
Temperature dependence of ξ^−2^ and *S*(0)^−1^ for linear-linear (L-L), ring-linear (R-L), and ring-ring (R-R) blends. The solid and dashed lines are linear fits to the experimental data and the arrows indicate the spinodal temperature, *T_s_*. Reproduced with permission from reference [137].

**Figure 15 polymers-12-01884-f015:**
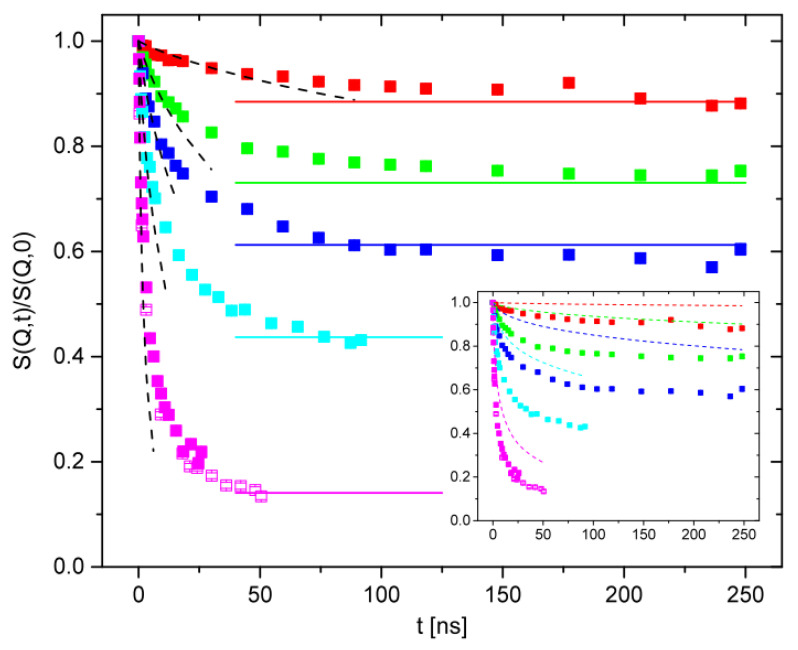
Intermediate scattering function for PEO rings (Φ = 0.1, *M*_w_ = 20,000 g mol^−1^) in a deuterated PEO linear matrix for *Q* = 0.05, 0.08, 0.1, 0.13, and 0.2 Å^−1^ (from top to bottom) at *T* = 413 K. Full and empty symbols refer to two different setups. The dashed black lines represent the initial Rouse decay. The solid lines indicate the plateau values with d = 42 ± 1 Å. The dashed colored lines correspond to the internal dynamics of pure rings. Reproduced with permission from reference [127].

**Table 1 polymers-12-01884-t001:** Molecular parameters for polymers discussed in solution or pure melts, in this study. Data are taken from reference [66], unless otherwise stated.

Polymer	Structure Repeat Unit	M_o_ ^(1)^/g mol^−1^	M_e_ ^(2)^/g mol^−1^	M_c_ ^(3)^/g mol^−1^
PDMS	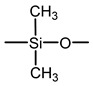	74.015	12,000	24,500
PEO	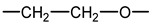	44.05	2000	5870
PS	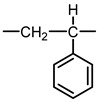	104.15	18,100	31,200
PE	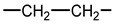	28.05	1150	3480

^(1)^ molar mass of the repeat unit; ^(2)^ entanglement molecular weight and ^(3)^ critical molecular weight at which entanglements effects are first observed.

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
