# Peer review of "Local Effects of Ring Topology Observed in Polymer Conformation and Dynamics by Neutron Scattering—A Review"

_polymers, 2020, doi:10.3390/polym12091884_

Round 1
Reviewer 1 Report
In this paper, the authors reviewed the researches related to neutron scattering analysis of cyclic polymers to summarize the physical properties of cyclic polymers. The manuscript extensively surveyed the research in the field of neutron scattering analysis of cyclic polymers, and the review will provide an indication to understand polymer behavior. However, the manuscript still has small insufficient and unclear description. The authors should revise the manuscript to improve the manuscript before publication in polymers.

Reviewer 2 Report
The current manuscript reviews conformation and dynamics of polymer melts and solutions as probed by neutron scattering experiments in cyclics and their linear counterparts. The text is clear and concise, with several important discussions, which I believe are of great interest for the audience reached by the journal, specially in the field of polymer physics. Based on the above, I recommend its publication, with minor changes, as stated below:
line 25, authors state that the review basically investigates PDMS, but several examples and discussions regarding PEO are also presented. So, such a statement should be broadened to situate the review better.
line 128, latest IUPAC recommendation is to use the term "dispersity" to replace the misleading, but widely used term “polydispersity index"
line 210, the solvent in which g-values for PDMS were obtained by Lutz and Higgins should be presented for comparison purposes. In the same sense, a comment about the effect of dispersity, molecular weight, concentration regime and solvent quality on the g-values, if data is available, could be included.
line 221, a standardization of terms is welcome in this section. Authors use 'undiluted state', 'melt state' and 'bulk', to, apparently, refer to the same state. Otherwise, the differences between the three states (if exist) should be clarified.
line 232, it is not clear for which dimensionality the d parameter refers.
line 277, data displayed in Figure 2 belongs to which polymer? The information is missing in the legend of the plot.
line 319, english editing needed, crumpled instead of crumped
line 323, it is not stated if data discussed in section 4 (Figure 5 and Table 1) regard the polymer melts or solutions. Please, specify this.
line 419, names should also be standardized here. Figure 6 claims "rings" and "chains", while the caption contains "cyclic" and "linear". Such a duplicity of terms is consistently seen in the whole text.
line 525, literature on polymer dynamics by NMR could also be referenced here.
line 562, "Q" symbol is missing in the caption of Figure 11.
line 1155, the journal name is missing in ref. 139.
